# Fourier Transform Infrared Spectroscopy Tracking of Fermentation of Oat and Pea Bases for Yoghurt-Type Products

Olivia Greulich [1,2], Lene Duedahl-Olesen [3,*], Mette Skau Mikkelsen [2], Jørn Smedsgaard [2] and Claus Heiner Bang-Berthelsen [1,*]

1   Research Group for Microbial Biotechnology and Biorefining, National Food Institute, Technical University Denmark, Henrik Dams Allè, Building 202, 2800 Kongens Lyngby, Denmark; oliviagreulich@gmail.com
2   FOSS Analytical A/S, Nils Foss Allé 1, 3400 Hilleroed, Denmark; mesm@foss.dk (M.S.M.)
3   Research Group for Analytical Food Chemistry, National Food Institute, Technical University Denmark, Henrik Dams Allé, Building 201, 2800 Kongens Lyngby, Denmark
*   Correspondence: lduo@food.dtu.dk (L.D.-O.); claban@food.dtu.dk (C.H.B.-B.); Tel.: +45-93518924 (L.D.-O.)

**Abstract:** The fermentation process of plant-based yoghurt (PBY)-like products must be followed for consistency by monitoring, e.g., the pH, temperature, and lactic acid concentration. Spectroscopy provides an efficient multivariate in situ quality monitoring method for tracking the process. Therefore, quality monitoring methods for pea- and oat-based yoghurt-like products using Fourier transform infrared (FT-IR) spectroscopy and high-performance liquid chromatography (HPLC) were developed and modeled. Plant-based yoghurt (PBY) was formulated by fermenting pea and oat plant drinks with a commercial starter culture based on *Lactobacillus* and *Streptococcus* strains. The main variance during fermentation was explained by spectral carbohydrate and protein bands with a notable shift in protein band peaks for the amide II band at 1548 cm$^{-1}$ to 1576 cm$^{-1}$. In addition to the identification of changed spectral bands during fermentation, FT-IR efficiently tracked the variation in oat and pea fermentation using pH as the main indicator. Prediction models with an $R^2$ for the predicted value of pH as a fermentation indicator ($R^2 = 0.941$) with a corresponding root-mean-squared error of prediction (RMSEP) of 0.247 was obtained when compared to the traditional pH method.

**Keywords:** plant-based; dairy alternatives; lactic acid; rapid method; chemometrics; principal component analysis (PCA); prediction; protein; pH





## 1. Introduction

The global demand for food must be met while minimizing the environmental impact of food systems. Plant-based (PB) meat and dairy alternatives, such as plant-based yoghurt-like products (PBYs), have gained increasing interest as a way to reduce the waste of proteins [1,2], minimize greenhouse gas emissions, meet dietary needs (e.g., intolerance or allergies), and a growing demand for dairy alternatives has been seen [3–5]. An annual growth rate of 11% for dairy alternatives in the European market [6] and a global market value for PB dairy alternatives estimated to reach USD 7.43 billion in 2027 [7] document this. Most raw materials for these dairy alternatives are based on crops like soy, rice, and almonds with a larger environmental footprint in production, water consumption, and transportation [8,9] than locally produced crops for fermentation. In the Nordic climate, such local raw materials include crops like oats and peas, which are the focus here. Few suitable cultures for fermented plant bases in comparison to dairy products have been reported for these purposes. Fermentation by lactic acid bacteria (LAB) has been proposed as a method of improving the flavor, nutritional value, and texture, and reducing the need for structuring agents in PBYs [10–13]. The fermentation of an oat drink with LAB showed an increase in amino acids, peptides, and phenolic acids and a decrease in antinutritional factors (ANFs); specifically, a reduction in ligans and phytic acid has been observed [14]. This illustrates the benefits of fermentation in PBY production, as it can create a higher

bioavailability of nutrients while reducing the number of antinutritional compounds such as phytic acid [14,15]. A challenge for plant-based alternatives is the large variability in the raw materials when compared to dairy milk.

Consequently, the demand for consistent quality in the industrial production of plant-based yoghurt (PBY) necessitates rapid and reliable fermentation monitoring methods. This requirement ensures not only the efficiency of production but also the maintenance of high-quality standards in the final products.

Quick at-line spectroscopic quality measurement methods enable the PBY industry to monitor the quality of the given batch within a few minutes during the fermentation. PBY products made from various protein raw materials constitute a broad range of products with different chemical compositions [16–23].

Spectroscopic techniques have long been established as valuable tools in the conventional dairy industry for monitoring various aspects of yoghurt production, from fermentation control to final product quality assurance. For instance, Arango et al. (2020) [20] introduced an NIR light backscatter sensor approach for in-line control during yoghurt fermentation, offering an alternative to cumbersome in-line pH measurements. Their study successfully established a pH prediction model with high determination coefficients (R2) (>0.993) and a low standard error of prediction (0.02–0.11 pH units), demonstrating the efficacy of spectroscopic methods in real-time process monitoring [20]. Final product analysis with NIR spectroscopy for the detection of protein adulteration in yoghurt has been employed by Xu and co-workers [23]. They utilized a one-class partial least squares model to compare pure yoghurt samples with potential adulterants such as edible gelatin, industrial gelatin, and soy protein powder at concentrations of as low as 1% ($w/w$) of edible gelatin [23].

These studies exemplify the versatility of spectroscopy throughout the yoghurt production value chain. However, while spectroscopic techniques have been extensively applied in conventional dairy yoghurt production, their utilization in the context of plant-based yoghurt (PBY) remains relatively underexplored in the literature with only a few studies. Among these, one study applied FT-IR spectroscopy for the stability of oat-based plant drinks [21], while another explored oat globulin conformations [22]. FT-IR signals are based on fundamental absorptions, making chemical interpretation easier compared to NIR, where signals are based on the overtones and combination bands of several chemical components [24]. Due to the higher specificity of FT-IR signals, it was the chosen spectroscopic method for this study, enabling an exploratory study of specific chemical components at discrete positions, and also allowing for the spectral study of secondary protein structure.

We, therefore, aim to address this gap by investigating the potential of Fourier transform infrared (FT-IR) spectroscopy as a rapid multivariate quality monitoring method for PBY throughout fermentation and in the final product stages. Specifically, the investigation focuses on locally produced protein sources such as oats and peas. Furthermore, we will compare the application of spectroscopy to conventional methods such as high-performance liquid chromatography (HPLC) where available. Through these investigations, we aim to highlight the unique and valuable contributions of this study, presenting a novel approach to PBY production.

## 2. Materials and Methods

### 2.1. Experimental Setup

Experiments were conducted in two steps with a preliminary fermentation study of several plant drink materials followed by a main fermentation study of the selected materials. The preliminary fermentation study was performed to screen various commercially available plant drinks based on local crops and the acidification dynamics of these by the inoculation of the commercial yoghurt culture. The selection criteria for the main fermentation included the requirement for observable acidification dynamics amenable to trend analysis, alongside a semi-homogenous final product with minimal phase separation

during fermentation so as to have a representative pH measurement. For an experimental overview, see Figure 1.

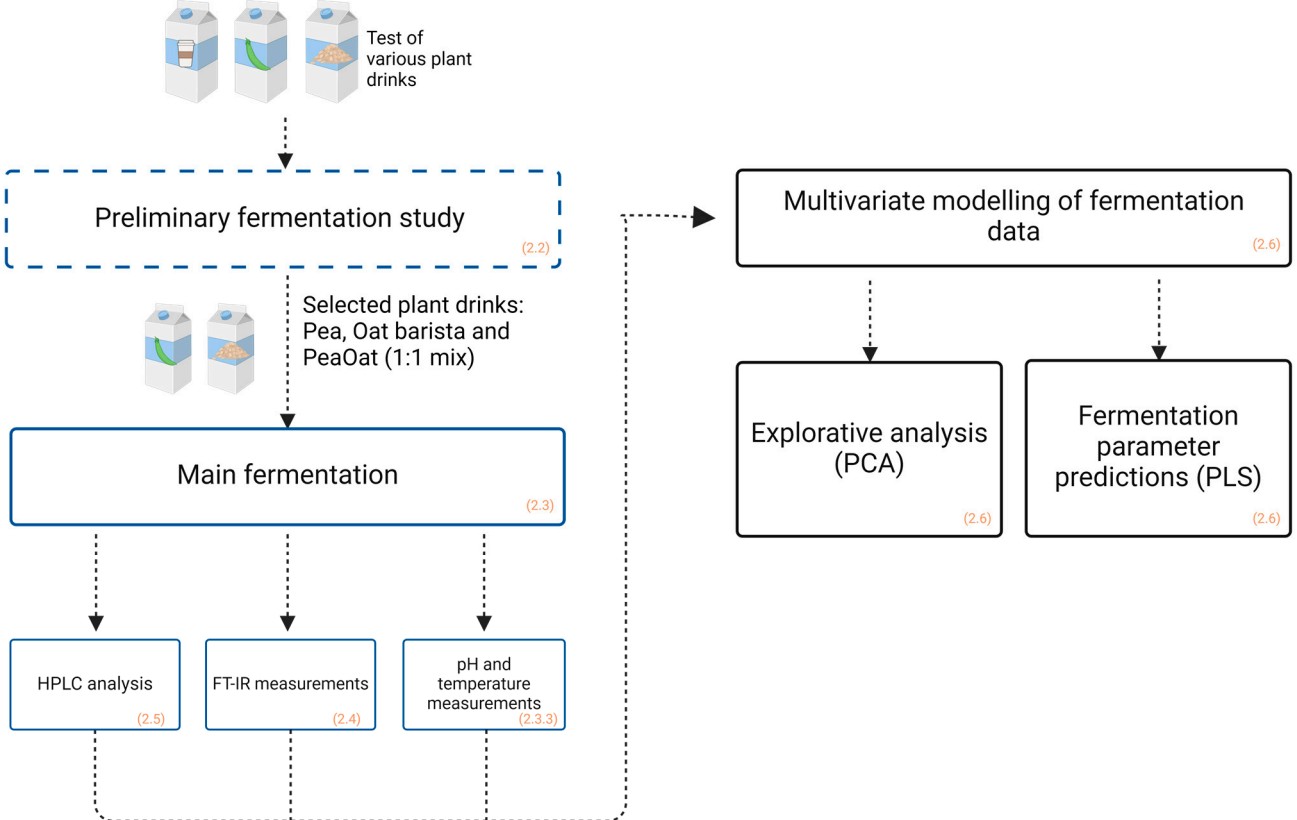

**Figure 1.** Overall experimental setup of the various analysis methods used in this study. Blue boxes indicate experimental parts and black boxes indicate data handling. Further details and descriptions can be found in the Material and Methods, specifically in the sections indicated by orange numbers in each box.

### 2.2. Preliminary Fermentation Study

In the preliminary fermentation study, commercial plant-based drinks (peas, oats, and mixes) were evaluated as potential fermentation media (Figure 1). Evaluated sample groups included a pea drink ("Ærte drik", DRYK Aps, Klippinge, Denmark), an oat drink ("Havre drik", DRYK Aps, Klippinge, Denmark), and a barista version of an oat drink ("Oat Barista Creamer", NATURLI' Foods A/S, Vejen, Denmark). Additionally, two 1:1 mixes were included in the assessment: one consisted of peas and regular oats, and the other of peas and the barista oat drink.

Three sample groups for further studies were selected, based on two commercial plant drinks: an oat barista drink (Oat barista creamer, Naturli'™, Denmark) and a pea drink (Ærte drik, DRYK™, Denmark). The sample groups are referred to as Pea, Oat, and PeaOat (1:1 pea and oat blend), each undergoing five replications, yielding a total of 15 independent fermentations. A mixture of Pea and Oat was included, due to previous reports on an improved protein composition, especially in relation to essential amino acids such as phenylalanine, leucin, and threonine [25].

### 2.3. Main Fermentation

#### 2.3.1. Sample Preparation and Inoculation

The fermentation of each sample type was performed in duplicate. Two autoclaved blue cap bottles were prepared, each containing 500 g of the sample-type mixture.

One bottle was dedicated to maintaining constant pH measurements, ensuring uninterrupted monitoring, while the other allowed for HPLC and FT-IR sampling throughout the fermentation process. The samples were inoculated with 0.02% (*w/w*) of commercial yoghurt starter culture (VEGA™ Premium, Novonesis, France), a mix of *Streptococcus thermophilus* and *Lactobacillus delbrueckii* subsp. *bulgaricus* for the plant material, according to the manufacturer's guidelines. The bottles were shaken by hand for 10 s and placed in a water bath to initiate the fermentation process.

The composition of the two sample types differs in terms of the raw material protein source and the formulation of the products (Table 1). Especially carbohydrate profiles and protein content differed between the sample types. The amount of sugar present in the Oat drink (4.3 g/100 mL) is threefold greater in comparison to that in the Pea drink (1.5 g/mL), whereas the quantity of protein in Pea is twice that found in Oat.

**Table 1.** Labeled nutritional values of the plant-based drinks used for the main fermentation.

| Nutritional Value [100 mL$^{-1}$] | Pea (Dryk) | Oat (Naturli) |
|---|---|---|
| Fat | 2.3 g | 2.9 g |
| -Saturated fatty acids | 0.2 g | 0.3 g |
| Carbohydrates | 1.5 g | 9.7 g |
| -Sugars | 1.5 g | 4.3 g |
| Protein | 2 g | 0.9 g |
| Ingredients | Water, pea protein (2.5%), rapeseed oil, sugar, acidity regulator: di calcium-phosphate, carriers: (calcium carbonate, calcium phosphate), gluten-free oat oil, salt, vitamins (D3, riboflavin, and B12) | Water, oat (13%), rapeseed oil, acidity regulator (tricalcium phosphate), sea salt, stabilizator (gellan gum), vitamins (D, B2 (riboflavin), and B12). |

### 2.3.2. Main Fermentation Trials

Sample bottles were placed in the water bath (WPE 45, Memmert, Germany) at 42 °C with a shaking speed set to a medium shaking rate (60 RPM). Samples were fermented for two hours, during which the pH, temperature, and FT-IR spectra were measured every 5 min for the first hour and then every 15 min during the second hour, theoretically yielding 240 spectral samples, although 248 were acquired due to some sampling errors. Samples for the HPLC analysis were taken at four points in time: prior to the inoculation of the culture (referred to as zero samples), after 25 min of fermentation, after 50 min of fermentation, and at the end of the fermentation (120 min). Upon sampling, the blue cap bottle (500 mL) was removed from the water bath and flipped vertically 3 times prior to sampling.

### 2.3.3. pH and Temperature Measurements

In the first 3 fermentations, a portable pH meter (Portavo 902, Knick, Germany) was used to measure the pH of the samples. An alternative pH meter (PHM210 Standard pH Meter, Lyon, France) was employed for subsequent fermentations. Both pH meters were calibrated using three-point calibration. Temperatures were logged simultaneously with the pH probe.

### 2.3.4. Spiking Experiments

Analytes of interest for the fermentation (glucose, fructose, and lactic acid) were investigated in the PB zero samples of Pea and Oat to gain insight into the spectral nature of the analytes on FT-IR. Analytes were added in a range of 4 concentration levels expected for glucose (Millipore®, CAS:14431-43-7, Germany) and fructose (Millipore®, CAS:57-48-7, Belgium) (1.5, 3.0, 4.5, and 6.8%) and for lactic acid (VWR, CAS: 50-21-5, France) (0.15, 0.3, 0.6, and 1.5%). After the addition of analytes to the PB samples, they were mixed for 30 s. Samples were subsequently measured with FT-IR.

*2.4. FT-IR Measurements*

The spectroscopic FT-IR measurements were carried out on a MilkoScan FT3 (FOSS Analytical A/S, Hillerød, Denmark) in the wavenumber range of 900–5000 cm$^{-1}$, with a 2 cm$^{-1}$ spectral resolution summarizing 40 scans. Blanks (deionized water) were measured prior to each fermentation start, and additionally between measurements when needed. A cleaning program was performed after each measurement using the manufacturer's cleaning solution (FOSS Analytical A/S, Hillerød, Denmark). The sampling of 30 mL in 50 mL cylindrical plastic sample tubes (FOSS Analytical A/S, Hillerød, Denmark) was performed as described in Section 2.3.2.

*2.5. Sampling and HPLC Analysis*

Upon the HPLC sampling points, 500 µL of PBY was collected and diluted 1:3 with autoclaved water. Samples were filtered through a 0.22 µm sterile filter (Labsolute, Th. Geyer, 7699822, Germany) into 2.5 mL Eppendorf tubes and put on dry ice. The HPLC samples were stored at −50 °C. The samples were thawed prior to analysis at 4 °C. The quantification of the lactic acid was conducted using an UltiMate HPLC (Dionex, Hvidovre, Denmark) based on calibration curves from 0.5 to 20 g/L. The analyses were completed on an Aminex HPX-87H ion exchange column ($300 \times 7.8$, pore diameter of 0.01 µm, BioRad) at 60 °C using isocratic elution (5 mM $H_2SO_4$) at 0.5 mL/min followed by refractive index detection (Shodex RI-101, Showa Denko K.K., Tokyo, Japan). Lactic acid was identified based on the retention time for analytically pure standards (Sigma Aldrich, USA) in 5 mM of $H_2SO_4$ as described previously [26]. Chromeleon 2.0 software (ThermoFisher, Boston, MA, USA) was used for visualization and quantification.

*2.6. Multivariate Data Analysis*

The data analysis was performed in Matlab 2020b (The MathWorks Inc., USA) using PLS Toolbox 8.9 (EigenVector Research Inc, USA). Noisy or non-informative spectral regions were observed and removed; hence, only the variables from the ranges 960–1586, 1728–1770, and 2836–2952 cm$^{-1}$ were included. The spectra were pre-processed with mean centering (MC) and the smoothing principle of Savitszky–Golay [27] (2nd order, 2nd derivative, and a window size of 15). See [28] for pre-processing strategies of spectral data. Initial predictions were made using all variables (approximately 750). Subsequently, outliers were identified and removed based on PCA analysis, utilizing Q residuals and T2 Hotelling as selection criteria. The model was cross-validated (5 splits) by replicate iterations to avoid bias in the model. To evaluate the performance of the partial least square, PLS models, root-mean-squared error (RMSE) values for the calibrated (RMSEC) and cross-validated (RMSECV) PLS models were used. Moreover, the ratio between RMSEC/RMSECV values was considered an indication of overfitting; thus, it was used in the choice of latent variables (LVs). $R^2$ was another measure of evaluating and comparing the PLS models' goodness-of-fit [16,29].

## 3. Results and Discussion

### 3.1. Acidification Dynamics

A higher variation between fermentations was observed for the Pea and PeaOat samples compared to Oat alone. The Pea samples exhibit a comparatively smaller reduction in pH during the initial 25 min of fermentation, decreasing from $7.7 \pm 0.14$ to $6.9 \pm 0.16$ pH, as compared to a significantly ($p < 0.05$) greater drop for Oat from $6.8 \pm 0.16$ to $5.1 \pm 0.2$ (Figure 2). This indicates that the Pea drink has a higher buffering capacity than the Oat drink, which could be attributed to the higher protein content of the Pea drink. Although the nature of the raw materials could also affect the buffering capacity as indicated by Wang et al., 2023 [30], who found that an oat drink with the same protein levels as a soy drink had a larger pH drop compared to soy, this could be caused by the lower buffering capacity of the oat drink due to the entire composition that includes other small constituents

contributing to the buffering capacity such as inorganic phosphate (e.g., from phytate), citrate, or organic acids [31].

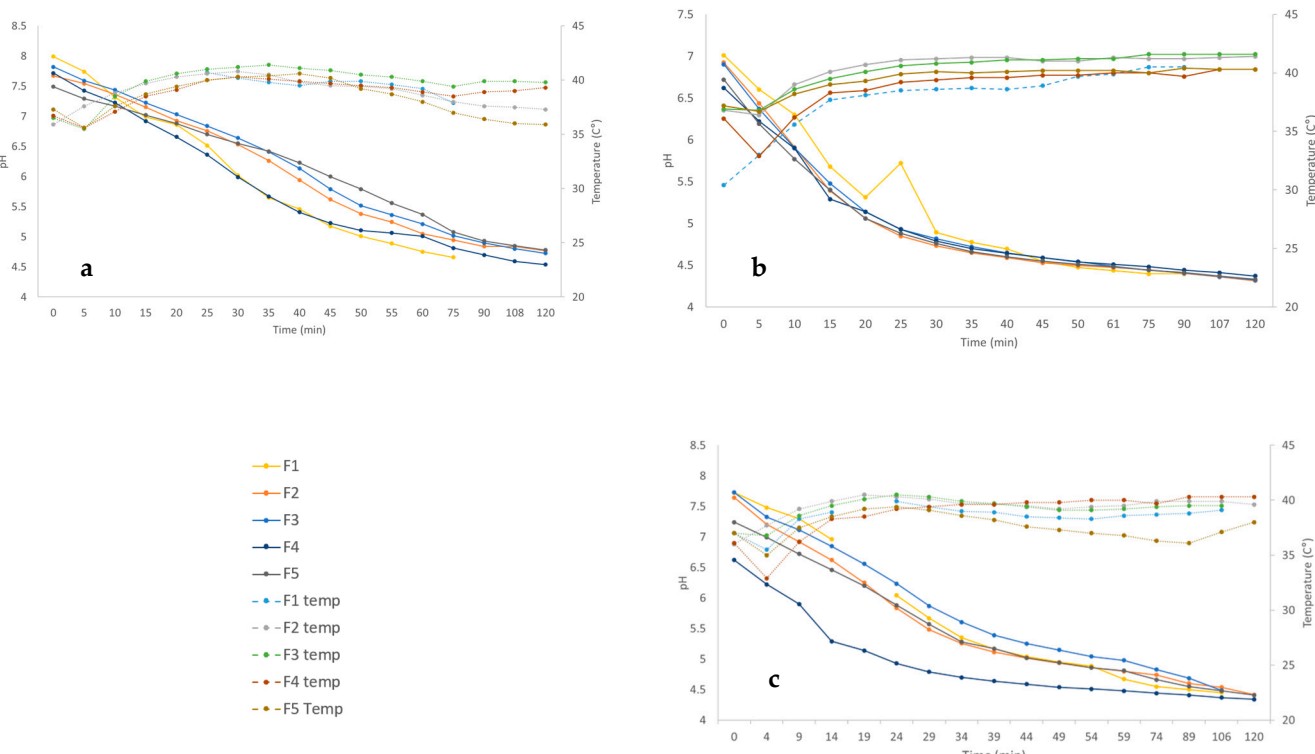

**Figure 2.** Acidification dynamics for Pea (**a**), Oat (**b**), and PeaOat (**c**), which include measured pH values (solid lines) and temperatures (dotted lines) from 5 fermentations (F1–F5).

In addition, the Oat drink exhibited a small variation between fermentations ($\pm 0.109$), which is attributed to the increased formulation of the Barista Oat drink with a higher fat content (rapeseed oil) and added stabilizers (gellan gum). The addition of thickeners in fermented dairy products has been reported to affect not only the physical stability of the product but also the sensorial experience, as previously described in the literature [32].

### 3.2. HPLC Quantification of Lactic Acid

The HPLC analysis was performed to obtain specific chemical information to predict key quality parameters such as lactic acid from the spectral data by PLS maximizing the covariance between the dependent variables (HPLC quantification) and the predictor variables (spectral data). The results from the HPLC analyses demonstrated that, similar to the pH results, the oat samples underwent rapid acidification during the first 25 min of fermentation. Subsequently, the concentration of lactic acid in the Oat samples plateaued at $1.77 \pm 0.34$ (g/L) and increased slowly to $2.47 \pm 0.33$ (g/L) at 120 min of fermentation (Figure 3). The Pea and PeaOat samples did not exhibit the same plateau development in the production of lactic acid as the Oat sample (Figure 3). The Pea samples exhibited the largest production of lactic acid with $5.16 \pm 1.55$ (g/L) after 120 min of fermentation, although it was also the sample type with the largest variation between samples at the same point in time as indicated by the standard deviations (Figure 3).

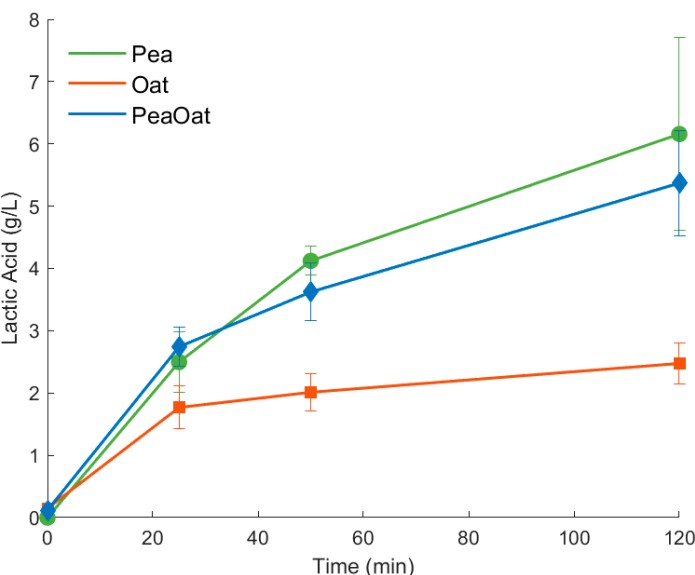

**Figure 3.** Lactic acid concentrations (g/L) measured by HPLC and refractive index for sampling points in time (0, 24, 48, and 120 h) during the fermentation of Oat, Pea, and PeaOat (1:1) samples.

In a recent study characterizing 25 different commercially available plant-based yoghurts based on soy, oat, coconut, or lupin, a similar lactic acid range between 0.31 and 7.77 g/L was recorded [33]. For oat-based yoghurt without acidity regulators, the lactic acid range was 1.25–2.32 g/L, whereas the other three yoghurt types contained between 3.5 and 6.2 g/L of lactic acid [33]. The lower lactic acid values of oat-based yoghurts can be related to the lower buffering capacity of the oat drink, which causes a reduced LAB activity compared to other plant sources such as peas or soy, similar to the findings of Wang et al., 2023 [30]. Studies have previously associated fermentation or added thickeners to a lupin-based dairy analogue with consumer acceptance [32]. Thus, the fermentation base should have a composition with an adequate buffering capacity, to develop an attractive texture and, hence, consumer acceptance. Another study investigated the consumer acceptance of protein-enriched oat-based gels and associated sensory attributes such as sweetness, moistness, softness, and smoothness as the main drivers of consumers liking the oat-based gels. Conversely, attributes such as sourness, chalkiness, and flouriness were found to detract from overall likability [3].

An enhanced understanding of consumer acceptance attributes equips producers with the opportunity to tailor products accordingly. In such scenarios, FT-IR-based rapid lactic acid measurements are proposed to facilitate the swift optimization of product analysis and modifications, thereby enhancing efficiency in product development and refinement processes.

### 3.3. FT-IR Spectral Data Variation

The raw spectral data from the main fermentation trials can be seen in Figure 4, where outliers and non-informative spectral regions have been removed (full spectra are found in Supplementary Figure S1). Outliers were of a spectral nature and clearly deviated from the remaining data with negative intensities throughout the entire spectra. The spectral outliers could be related to the heterogeneity of the samples or whether the instrument was not sufficiently cleaned between measurements. Sample types were grouped together, except for a slight deviation in PeaOat and Pea, where the intensity variation was higher within each sample type, indicated by peaks with varying intensities at specific wavenumbers. The compositional differences of Pea and PeaOat samples containing more protein compared to Oat, and the Oat samples having higher carbohydrate content compared to Pea (Table 1) were attributed to being parts of the variation observed in the spectra (Figure 4).

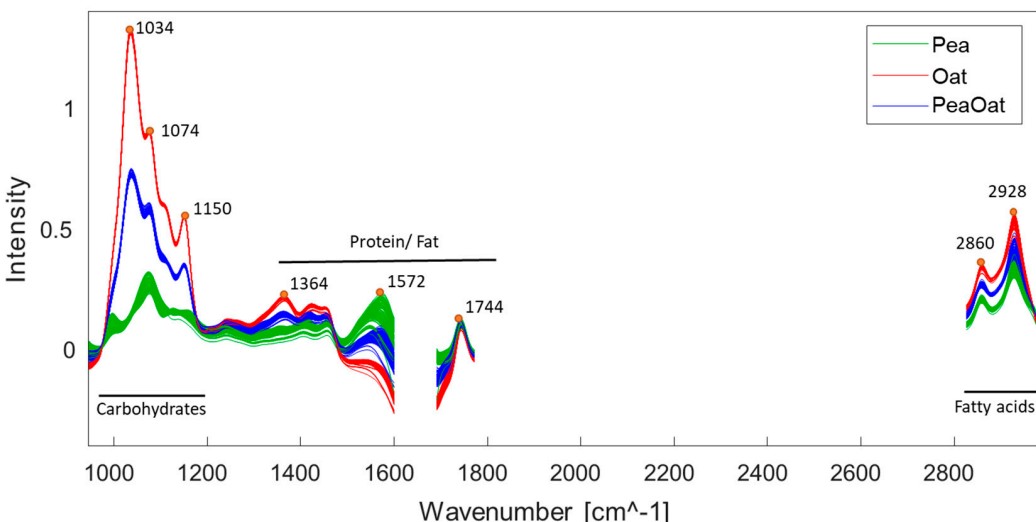

**Figure 4.** Raw FT-IR absorbance spectra from the 5 fermentation trials colored by sample type. Five spectral outliers and non-informative spectral regions were removed and resulted in plotted regions of 960–1586, 1728–1770, and 2836–2952 cm$^{-1}$.

Upon the investigation of the variation in wavenumber regions between sample types in Figure 4, a high variability is observed in the carbohydrate fingerprint region (1200–900 cm$^{-1}$). Peaks in this region were at 1034, 1074, and 1150 cm$^{-1}$ (Figure 4), corresponding to C-O stretching absorbance, which are connected to the pyranose form of the carbohydrates (e.g., glucose and starch) [34,35], were also seen to be dominating in Oat (Figure 4). More specifically, the peak at 1074 cm$^{-1}$ was previously connected to the stretching of a C-O ether bond [36] and similar peaks in oat-based drinks were previously reported at 1024 and 1151 cm$^{-1}$ [21]. An absorption band at 998 cm$^{-1}$ has previously been reported to be due to sucrose [37], which corresponds well with the HPLC data (not shown) for Pea, which, according to Table 1, also contained added sugar (expected to be sucrose).

Fatty acid peaks are mainly found around 2960, 2929, and 1740 cm$^{-1}$ [38]. In Figure 4, peaks are seen at 2928 and 2860 cm$^{-1}$, corresponding to asymmetric and symmetric CH$_2$ stretching modes. The observed peak at 1744 cm$^{-1}$ corresponds to the stretching of C=O ester groups in triglycerides [21,39].

Protein is usually found by the absorbance of amide I (1600–1700 cm$^{-1}$) due to the stretching of the C=O bond of the amide; of amide II (1510–1570 cm$^{-1}$) due to the bending vibration of the N-H bonds; and of amide III (1350–1200 cm$^{-1}$) due to the combination of the plane bending of N-H and C-N stretching [40]. Within these regions, peaks are observed in Figure 4 at 1572 and 1364 cm$^{-1}$ for amide II and amide III, respectively. Peaks for amide I are not detected as the variables are excluded due to the too high signal-to-noise ratio. The high sensitivity of the amide I band has been used to explain secondary protein structures, although the use of further deconvolution methods is needed to separate the peaks [41].

### 3.4. Spectral Development during Fermentation

The largest variation within the sample types was observed in the Pea samples, wherefore the spectra of Pea zero samples were compared to the spectra of Pea samples from the end of fermentation to examine the time dependence of this variation (Figure 5). The Pea zero samples spiked with glucose, fructose, and lactic acid were included as references to identify the signals corresponding to these compounds. The choice of compounds was made according to analytes present in the raw spectral data analysis as well as the unpublished HPLC results of raw and fermented materials, indicating differences in concentrations for these compounds during fermentation.

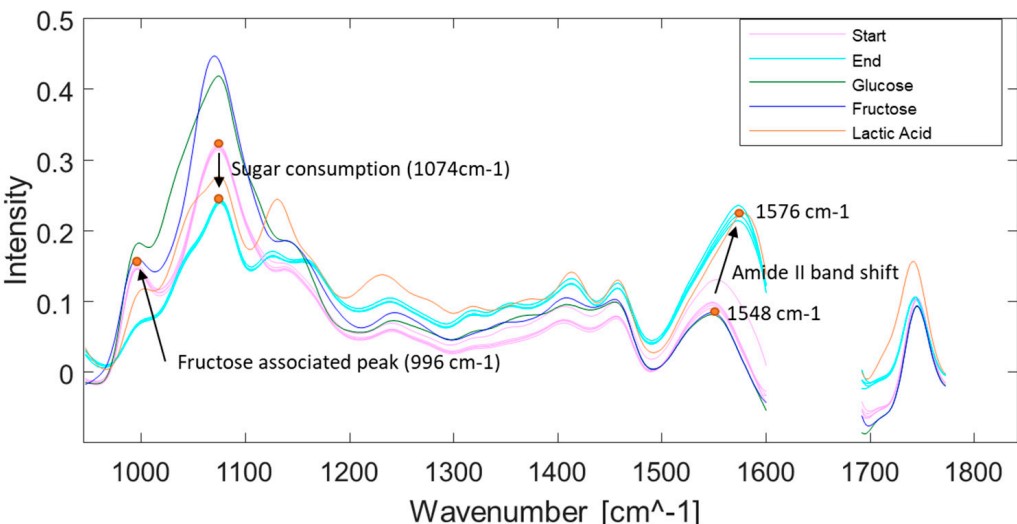

**Figure 5.** Spectra of Pea zero sample (start) and at the end of fermentation (end), including spectra of spiked Pea zero samples (start samples) with glucose (2.72 g/100 mL), fructose (2.70 g/100 mL), and lactic acid (1.53 g/100 mL).

A notable difference was observed in the carbohydrate fingerprint region between the start and end samples, both in terms of the decrease in the curve and the changes in peak shape resulting from sugar consumption during fermentation. Previous reports have stated that *Lactobacillus delbruekii* subsp. *bulgaricus* is not able to ferment sucrose, whereas *Streptococcus thermophilus* converted sucrose into acids like lactic acid [42], which is believed to be part of the acidification seen here. The formed acids and carbon dioxide have been reported to create a growth-simulating environment for *Lactobacillus delbruekii* subsp. *bulgaricus* [25]. Degradation of the disaccharide sucrose results in the formation of glucose and fructose, which could explain the low decrease and change in peak shape in the carbohydrate region signals during fermentation (Figure 5; pink and light blue). The highest peak and, therefore, the spectral maximum, was observed at 1074 cm$^{-1}$, where the start samples had a similar shape to the fructose-spiked samples (dark blue) and the end samples had a similar shape to the glucose-spiked samples (dark green). A small shoulder was also observed at 996 cm$^{-1}$. Comparing a 1.5% solution of fructose and glucose diluted in water to the spiked samples revealed that the shoulder peak at 996 cm$^{-1}$ (Figure 5) was only present for fructose and not for glucose, while the same peak was also previously reported to represent sucrose, as stated by Lendl [37].

An increase in intensity and a slight shift in the peak are observed for the amide II band from 1548 cm$^{-1}$ at the start of fermentation to 1576 cm$^{-1}$ at the end of fermentation (Figure 5). This may be attributed to the change in secondary protein structure during fermentation. The proteolytic *Lactobacillus delbrueckii* subsp. *bulgaricus* contributes to this change by degrading proteins to peptides and free amino acids, which function as sources of nitrogen for *Streptococcus thermophilus* (Bonke et al., 2020) [25]. A synergistic effect that, together with the s synergy explained for carbohydrates, has also been shown for soy fermentation [43,44] The shift of peaks could, however, be related to the change in the β-sheet structure of the proteins from antiparallel to parallel [45]. Similar bands were reported for fermented faba bean blends, where amide I and II bands were observed at 1632 cm$^{-1}$ and 1538 cm$^{-1}$ [46]. In their study, they did not find any shift in the amide II band during fermentation but did observe a relative decrease in the content of β-sheet (1633 cm$^{-1}$) and an increase in random coil structure (1644 cm$^{-1}$) during fermentation [46]. Random coil was attributed to the disrupted structure of the beta-sheet during fermentation. Furthermore, it was suggested that hydrogen bonds were disrupted enough during fermentation to partially unfold the protein structure. Similar interpretations can most likely be applied to this study. Comparable observations were made by others in a study on the fermentation of whey

with blueberry juice using *Lactobacuíllus casei*, where changes in secondary protein structure were observed with reduced intensities across the protein bands during fermentation [47]. Identical observations were seen for fermented soybean meal compared to unfermented soybean meal [48].

The results of the HPLC analysis showed that, in accordance with the pH measurements, the Oat samples underwent rapid acidification during the first 25 min of fermentation (Figure 3). The lactic acid concentration of the Oat samples remained constant, while the Pea and PeaOat sample concentrations increased to levels above 5 g/L. On FT-IR, the presence of a peak in the same region for the lactic acid-spiked samples (Figure 5) may be due to changes in the protein structure resulting from the change in pH (e.g., acid hydrolysis). However, the secondary protein structure and solubility could also be related to other factors rather than pH alone. Other studies observed a significantly lower solubility of fermented pea protein isolate at neutral pH compared to unfermented pea protein isolate [49]. In addition, observations of slight protein precipitation after spiking suggest that a direct relationship between the absorbance of lactic acid molecular bands cannot be conclusively established. Further investigations with targeted enzymes should be carried out to fully conclude the protein structure development observed in the spectra.

*3.5. Investigating Captured Variance by PCA*

To investigate how much variance is explained by the data, a principal component analysis (PCA) was performed (Figure 6). PC1 explains 88.47% of the variance and clearly separates the sample types into raw material categories. The loadings for PC1 (Figure 6c) indicate that PC1 is mainly explained by the carbohydrate fingerprint region and the asymmetric and symmetric $CH_2$ stretching modes of fatty acids at $>2800$ $cm^{-1}$. The variance in the carbohydrate profile could be explained by the added sugars in the Pea samples in contrast to the Oat samples, which had no added sugars. PC2 explains a 10.72% variance, where loadings indicate that some of the carbohydrate information is partly explained by a sharp glucose peak at 1030 $cm^{-1}$ (Figure 6c). However, the model for predicting glucose had poor performance, with an $R^2(pred) = 0.557$ and RMSEP = 0.086.

There is a positive, almost linear correlation between PC1 and PC3 for the Pea samples (Figure 6a). This correlation is pH-dependent, which does not apply to the Oat and PeaOat samples to the same extent. PC3 only explains a 0.66% variance, where high scores on PC3 are acidic (Figure 6b). Loadings for PC3 show that the variance explained is mainly contributed by the amide II band (N-H bending) at 1582 $cm^{-1}$ and a sharp peak at 1034 $cm^{-1}$ in the C-O stretching region. The combination of the carbonyl groups of sugars and the N-H bonds could indicate some potential glycosylated protein as a result of LAB-induced glycolysis as described by others (Welman et al., 2003; Montemurro et al., 2003 [4,50]). In a study of the microwave glycation of soy protein, FT-IR spectroscopy showed a similarly increased absorbance in the C-O stretching region and O-H deformation vibrations at 1100–1050 $cm^{-1}$ in combination with a decrease in NH2 bands during microwave glycation of soy protein [51].

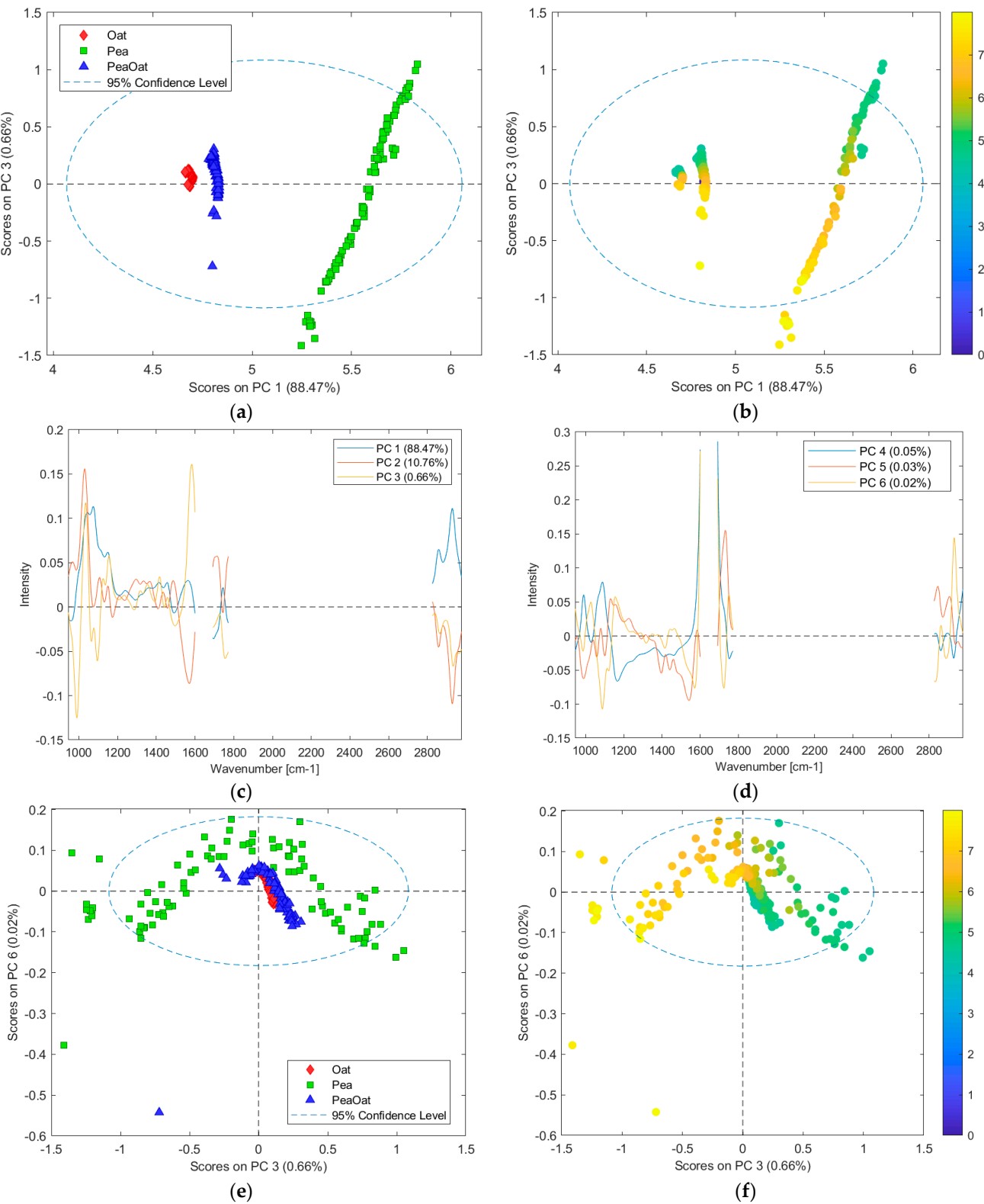

**Figure 6.** PCA score plots of components 1, 3, and 6, spanning relevant variations within the samples (Oat, Pea, or PeaOat (1:1)), where (**a**,**e**) are colored by sample type and (**b**,**f**) are colored by pH. The loadings of the 6 principal components (PCs) of the PCA model are shown in (**c**,**d**), n = 248.

The remaining PCs were investigated to search for other variations during fermentation that were more pH-dependent. PC6 only explained 0.02%, but when combined with PC3, an interesting trend was observed (Figure 6f). The Pea samples were explained to a higher degree by PC3 compared to Oat and PeaOat (Figure 6e), especially on the negative score of PC3. At alkaline pH, a positive correlation between PC3 and PC6 was observed, while a negative correlation was observed for acidic pH. The PC6 loadings in Figure 6d showed a slight amide band shift ($+20$ cm$^{-1}$) from the amide II peak of the PC3 loading. Similar observations were made in the raw spectral data during the fermentation of the pea samples (Figure 4). Similar findings were reported by Patra [21] for oat-based drinks, although the calibration data included the variable range between 1600–1700 cm$^{-1}$. This effect could be connected to the protein solubility change at a pH below the isoelectric point of the proteins caused by the *S. thermophilus* conversion of sucrose into acids like lactic acid [42]. A study found a minimum solubility of pH shift-treated pea protein isolate around pH 4–5 [52], which resonates with the scores on PC3, where score values close to 0 (center in Figure 6b) are at pH 4, resembling the maximum insolubility of the pea proteins.

In the industrial production of PBY, oat-based fermentation demonstrates rapid acidification, making it an efficient option for swift fermentation processes. However, while oat media offer rapid acidification, they may not yield the same textural qualities as their pea-based counterparts. In comparison to oats, fermented pea drinks exhibit a more viscous texture, likely due to their elevated protein content, as observed by visual inspection (data not included). To monitor protein development, a PCA analysis was applied in this study; nevertheless, further investigation into protein solubility and glycolysis is recommended. This could be achieved through methods such as enzymatic exposure to proteases and glycolysis-inducing enzymes such as hexokinase or lactate dehydrogenase. Such methods could provide valuable insights into the fermentation process and product characteristics.

*3.6. Prediction of pH by Partial Least Squares*

Prediction models for acidification parameters, pH and lactic acid, were developed to test the feasibility of FT-IR as a monitoring method during the fermentation of PBY. The final optimized pH prediction model performed with an RMSEP = 0.247 and predicted $R^2$ = 0.941 (Figure 7). The removal of spectral outliers improved the model performance by an $R^2$ of 0.143 and a reduction of 0.222 in RMSECV. Similar pH predictions have been made with NIR for thermophilic solid-state fermented soybean meal with a predicted $R^2$ = 0.98 and an RMSEP of 0.169 [29]. Another study for real-time pH monitoring with NIR during UHT milk fermentation predicted a pH decrease of 5.2–4.6 with an $R^2 > 0.993$ and an RMSEP between 0.02 and 0.11 pH units [20]. FT-IR pH prediction models for pH have been evaluated for a mulberry application with an $R^2$ = 0.97 and an MSE = 0.0003 [53]. This indicates that such modeling techniques for fermentation monitoring are feasible with NIR and FT-IR spectroscopy, although the RMSE values for soybeans reported by others and the materials included in this study do not perform like the UHT milk and mulberry pH predictions. The Hotelling values for the measured pH in the given study indicate the variation in the samples within the model, where pH values above or below 6.5 $+/-$ 0.5 are observed to have a larger variation within the model, which could be related to the stability of the pH meter at pH measurements above or below neutral pH. In addition, the repeatability of each fermentation trial had some temperature and pH variations during fermentation (Figure 2), which is expected to affect the prediction accuracy.

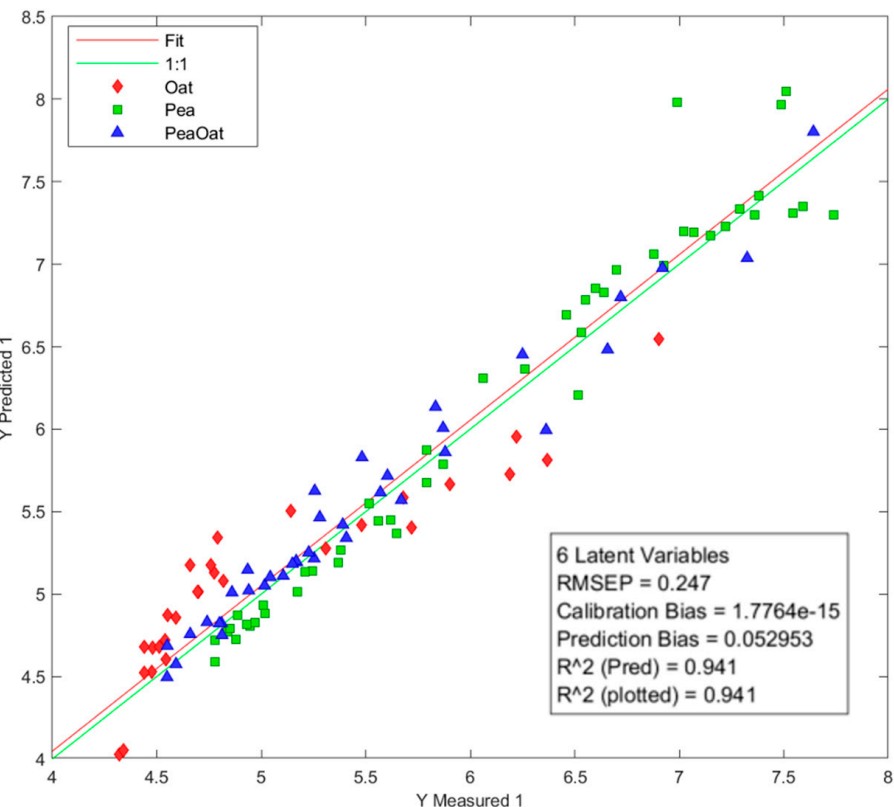

**Figure 7.** Linear fit of measured vs. predicted pH plot of six latent variables (LV) colored according to sample type (Oat, Pea, and PeaOat). X block data split 70/30 into calibration and validation sets, respectively, n = 241. Model prediction outcome in the box with, e.g., R2 predicted = 0.941.

The main explained variance (98.26%) of LV1 (Figure 8d) primarily explains the difference between the sample types, as seen in the PCA (Figure 6a). LV2 (Figure 8e) with a 1.42% explained variance appears to describe a decreasing or increasing tendency as a function of pH. The Oat samples form a clear sigmoid curve, with Pea and PeaOat following a more linear curve with a light sigmoid trend prior to and post-pH 6 similar to the results shown in Figure 2. The sigmoid trend could represent bacterial growth phases, as described by Parvarei et al., 2021 [54], who observed five phases during the fermentation of milk, including a lag phase, pre-log phase, log phase, late phase, and stationary phase [54]. Similar phase trends are observed during the oat drink starting from the initial pH where long lag and pre-log phases are observed, followed by a high log slope; the following late and stationary phases are not equally represented. This could be due to the lower buffering capacity of the Oat compared to the Pea and PeaOat samples as seen in the pH measurements (Figure 2). Previous studies have shown a similar trend of a fast pH drop during oat fermentation, which was similarly speculated to be caused by the lower buffering capacity of the oat drink compared to a soy drink [30]. The pH model accuracy was observed to be lower for samples above pH 7, which was supported by the Hotelling values (Figure 8c). This indicated a lower performance of the model during the lag phase of fermentation. Based on loadings, slope trends, and observations, we consider the variance by LV2 to express fermentation pH.

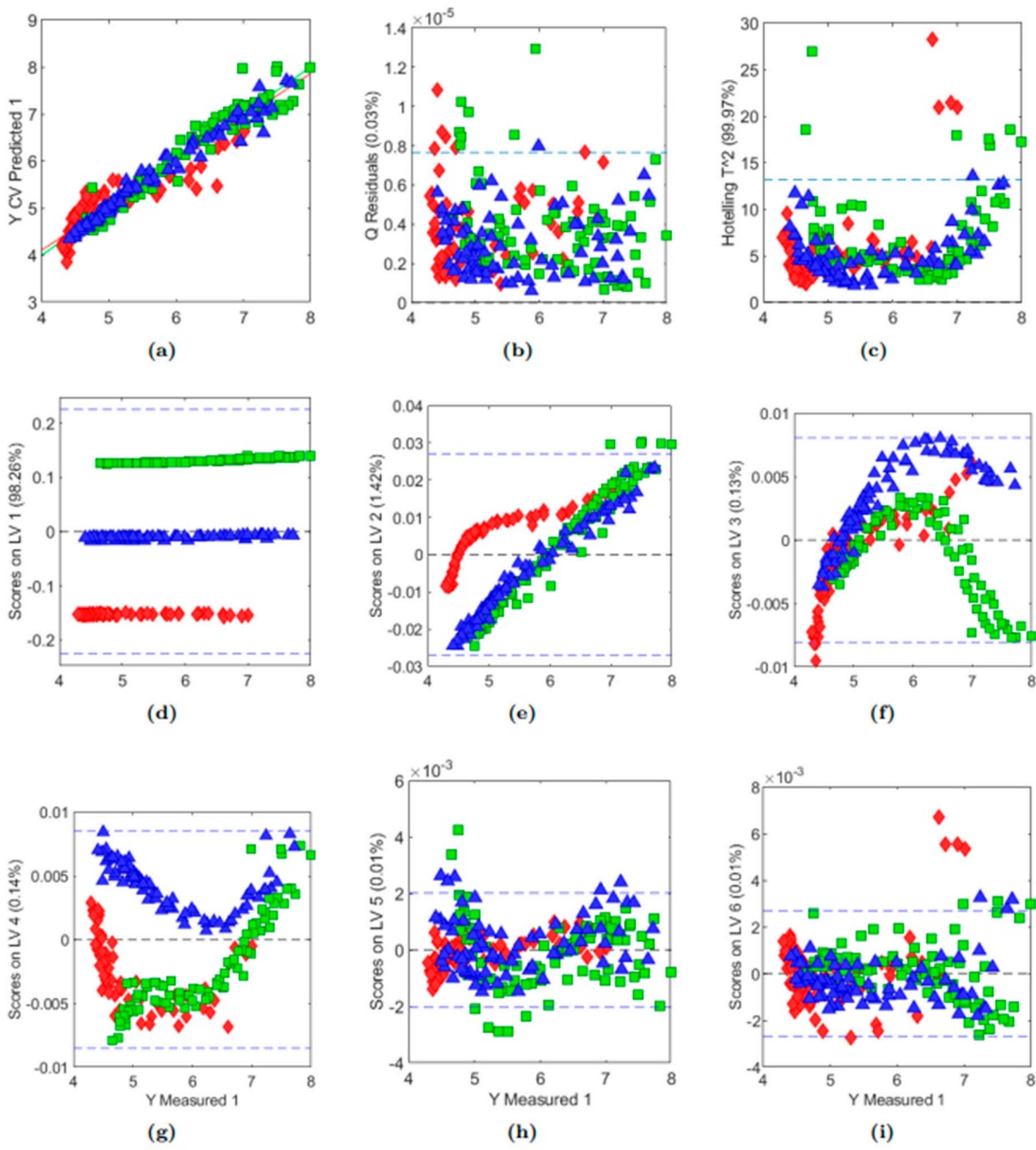

**Figure 8.** Plots from the modeling of the pH prediction model with Y measured (pH) as the horizontal axis for all plots. Vertical axis for (**a**) Y predicted values vs. measured Y (pH) with an $R^2$ = 0.92 (CV). Plot (**b**) Q residuals for measured pH, (**c**) Hotelling $T^2$ values for measured pH; plots (**d**–**i**) are individual latent variable (LV) scores for the measured pH, respectively. Red diamond shapes are Oat samples, blue triangles are PeaOat, and green squares refer to Pea samples.

An interesting inverted U-shaped or even sinusoidal curve is observed for the scores of LV3 (0.13% explained variance) (Figure 8f). The Pea samples show an increase until around pH = 6, after which a drop in the scores is observed. For the Oat samples, a sinusoidal curve is observed with a large drop towards acidic pH. Loadings show that LV3 is mainly explained by three peaks (Supplementary Materials Figure S2). Two peaks are shown at 1110 and 1150 cm$^{-1}$, which correspond to the two shoulder peaks observed for the spectra of glucose diluted in water. Furthermore, the loadings show that a high peak at 1572 cm$^{-1}$,

which has been related to protein content by others [40], also explains LV3. The explained variance could be attributed to the solubility changes of the glycosylated protein at alkaline and acidic pH compared to the variance at the pI of the protein. This would align with the larger variance observed for the Pea samples, which have a higher protein content compared to Oat (Table 1). Similarly, Jiang et al., 2022 [52] reported a decrease in the solubility of pea protein around pH 4–5 (the protein's pI), alongside an expanded pH range of solubility following its conjugation with inulin. The conjugation was overall shown to improve the functionality of the pea protein isolate, which was partly attributed to increased glycosylation. A previous study by Khajehpour and coworkers [55] presented a semi-quantitative method for measuring glycosylation based on the absorbances of the IR fingerprint region (1000–1200 cm$^{-1}$) for sugars and amide bands, typically observed around 1566 cm$^{-1}$. The study did not propose the method as a quantification replacement but as a tool for the kinetic monitoring of glycosylation processes during fermentation. Glycosylation has previously been observed to increase the thermal stability of proteins [56,57].

The scores for LV4 showed a U-shaped curve for the PeaOat and Oat samples and a sinusoidal curve for the Pea samples during the fermentation process (Figure 8g). The LV4 scores exhibited a decreasing trend until pH 6, after which the Pea sample scores remained relatively constant until pH 5, at which point a further decrease was seen (Figure 8g). In contrast, the scores for the Oat and PeaOat samples increased to their original values at the beginning of the fermentation.

An analysis of the loadings revealed that the explained variance in LV4 was likely due to the carbohydrate profile of the samples, with peaks at 978 and 1044 cm$^{-1}$ corresponding to sucrose, fructose, and glucose (Figure 5). The LV4 loadings also indicated that the explained variance might be related to the fatty acid profile, with peaks at 1586, 2870, and 2944 cm$^{-1}$. A peak at 1764 cm$^{-1}$ was associated with the protein content. The trend in the LV4 scores was noteworthy, although the loadings were somewhat difficult to interpret. It is possible that the LV4 scores reflect the variance in carbohydrate consumption during the fermentation process, specifically the consumption of reducing sugars. More investigation is required to validate these results.

### 3.7. Prediction of Lactic Acid by Partial Least Squares

Lactic acid production during the fermentation has previously been used to predict the end time of fermentation using a combination of a mechanistic model and an ANN model, where lactic acid, biomass, and lactose were used as the data input [58]. In our study, the prediction of lactic acid was investigated, and an optimal model was a reduced PLS model with four LVs (reduced from Figure 7). The calibration model consisted of 39 samples, where 4 outliers had been removed due to high T$^2$ Hotelling and Q residual values, thus improving the model in terms of RMSE and R$^2$. This removal caused a reduction in the representation of PeaOat samples. The validated model predicted lactic acid with an RMSEC = 0.045, RMSECV = 0.052, RMSEP = 0.049, and the predicted R$^2$ = 0.933 (Figure 9).

The better performance of lactic acid during the whole fermentation period indicates that the spectral data can model the acidification; however, more data are needed to validate this. In a practical context, the real-time multivariate fermentation monitoring facilitated by FT-IR enables operators to make informed decisions promptly, optimizing resource utilization and reducing production costs. In this study, emphasis was placed on pH and lactic acid as the key fermentation indicators, whereas future studies could investigate prediction models for protein solubility and batch-end quality monitoring for PBY production.

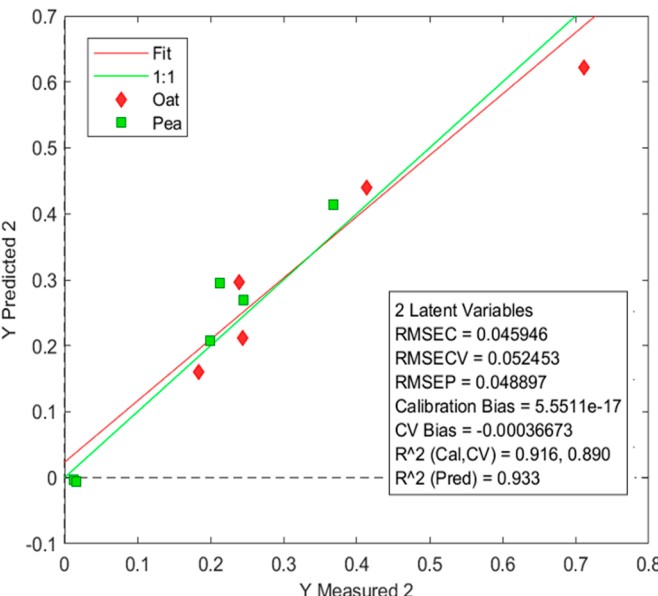

**Figure 9.** Measured vs. predicted lactic acid (g/L) plot for two latent variables (LV). Colors relate to Oat (red diamonds) and Pea (green squares). The linear calibration model was based on n = 39; the model prediction outcome is included in the box with, e.g., R2 predicted = 0.933.

## 4. Conclusions

The fermentation and quality monitoring capabilities of Fourier transform infrared (FT-IR) spectroscopy and high-performance liquid chromatography (HPLC) were evaluated for the fermentation of pea, oat, and mixed pea and oat drinks. The multivariate analysis of HPLC and spectral data was used to develop a rapid fermentation monitoring method using principal component analysis (PCA) and partial least squares (PLS). The FT-IR spectral data showed significant variation in the samples during fermentation, particularly in the carbohydrate fingerprint region (1200–900 cm$^{-1}$) and the amide II band (1572 cm$^{-1}$) in pea samples. The model for the main fermentation indicator, namely pH, was based on PLS loadings and had an R$^2$ (pred) = 0.941 and an RMSEP = 0.247, with LV2 speculated to represent the variation in pH. Furthermore, similar tendencies, as observed in PCA, of protein and carbohydrate variations during fermentation were observed in LV3 and LV4, respectively. FT-IR spectroscopy was able to track carbohydrate and protein variations during the fermentation of oat- and pea-based yoghurt using multivariate modeling methods such as PCA and PLS, offering a potential for a fast and reliable process and product monitoring for plant-based food manufacturers. Our data support the notion that future studies on protein development and solubility in plant bases will contribute to a better understanding of the final product quality.

**Supplementary Materials:** The following supporting information can be downloaded at https://www.mdpi.com/article/10.3390/fermentation10040189/s1: Figure S1: Raw spectra of all samples and variables of the main fermentation trial. Figure S2: All six loadings for pH PLS model on all data.

**Author Contributions:** Conceptualization, L.D.-O., M.S.M., J.S. and C.H.B.-B.; Data Curation, O.G.; Formal Analysis, O.G.; Funding Acquisition, C.H.B.-B.; Investigation, O.G.; Project Administration, C.H.B.-B.; Supervision, L.D.-O., M.S.M. and J.S.; Validation, O.G. and M.S.M.; Writing—Original Draft, O.G.; Writing—Review and Editing, L.D.-O., M.S.M., J.S. and C.H.B.-B. All authors have read and agreed to the published version of the manuscript.

**Funding:** This project has received funding from Innomission 3 partnership AgriFoodTure, funded by Innovation Fund Denmark and the European Union NextGenerationEU, under grant agreement No 1152-00001B. This output reflects only the author's view, and the funding organization cannot be held responsible for any use that may be made of the information it contains. All authors are directly involved in the Innomission 3 REPLANTED project.

**Institutional Review Board Statement:** Not applicable.

**Informed Consent Statement:** Not applicable.

**Data Availability Statement:** Data and Supplementary Materials are contained within the article.

**Acknowledgments:** The authors would like to thank Novonesis for providing the VEGA^TM Premium culture and technical assistance. Graphical abstract and Figure 1 were created at BioRender.com (accessed on 31 December 2023).

**Conflicts of Interest:** All industry partners in REPLANTED: Novonesis, Thise Dairy, KMC Amba, and FOSS Analytical A/S approved the manuscript before submission for publication. M.S.M. and O.G. are employees of FOSS, which produces and sells FT-IR spectroscopy platforms. The remaining authors (J.S., L.D.-O., and C.H.B.-B.) have no competing interests to declare.

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
