# Peer review of "Fourier Transform Infrared Spectroscopy Tracking of Fermentation of Oat and Pea Bases for Yoghurt-Type Products"

_fermentation, doi:10.3390/fermentation10040189_

Round 1
Reviewer 1 Report
Comments and Suggestions for Authors
The manuscript effectively addresses the inherent variability in the fermentation process of plant-based yogurt (PBY) arising from differences in plant raw materials. The choice of Fourier transform infrared (FT-IR) spectroscopy and High-Performance-Liquid-Chromatography (HPLC) for quality monitoring is well-aligned with the study's objectives. The utilization of a commercial starter culture for fermenting pea and oat plant drinks enhances the practical relevance of the research. However, providing additional details about the specific starter culture employed would further enrich the study. The identification of spectral carbohydrate and protein bands as primary contributors to variance during fermentation constitutes a pivotal finding. The observed shift in protein band peaks is noteworthy, providing valuable insights. The manuscript convincingly illustrates the efficacy of FT-IR in tracking variations in PBY fermentation based on pH and lactic acid concentration. The incorporation of prediction models with R2 values and RMSEP adds quantitative rigor to the results, enhancing the overall robustness of the study. The comparison of FT-IR results with traditional pH and HPLC methods enhances the practical applicability of the proposed spectroscopic approach. While the text is generally clear and well-presented, some crucial issues require attention before acceptance in the esteemed Fermentation MDPI Journal.
#1 Every abstract should only contain relevant information that summarizes the work. Also adds quantitative relevant informations to the abstract if relevant. I recommend authors use the following reference to adjust their abstract (https://doi.org/10.1016/j.carbon.2007.07.009).
#2 There are several typographical and grammatical mistakes which should be corrected, e.g., the missing verb in "This illustrate the benefits of fermentation..." (should be "illustrates") and the phrase "explored protein adulteration using spectroscopy" (may benefit from clarification).
In introduction
#3 The transition between the global demand for food and the specific focus on PBY could be smoother. Clearly stating the relevance of PBY in addressing the challenges outlined at the beginning could enhance coherence.
#4 The connection between the benefits of fermentation mentioned earlier and the need for rapid monitoring methods is somewhat abrupt. Adding transitional phrases could improve the flow.
#5 While the introduction mentions various spectroscopic techniques, a brief explanation of why FT-IR spectroscopy was chosen over others would provide better context for readers unfamiliar with the field.
#6 Mentioning the estimated global market value for PB dairy alternatives is relevant, but providing a source for the estimation and specifying the relevance to the study would enhance credibility.
#7 The novelty of the work is not expressed explicate. In which aspect this work is original and better than others?
In Material and methods section
#8 While the section mentions a two-step process with a preliminary fermentation study followed by a main fermentation study, the exact rationale for these steps is not clear. Providing a brief explanation of why the study was conducted in two steps would enhance clarity.
#9 The section describes the preliminary fermentation study using commercial plant-based drinks (pea, oat, oat blends, and mixes). More information about the specific brands, formulations, and nutritional content of these drinks would be beneficial for the reader to understand the experimental conditions.
#10 The criteria for selecting the three sample groups (Pea, Oat, and PeaOat) are briefly mentioned, but additional details on the selection process or specific characteristics that influenced the choice would enhance the reader's understanding.
#11 The section on main fermentation provides clear details on sample preparation, inoculation, and the fermentation process. However, it would be helpful to explain why two autoclaved blue cap bottles were used for each sample type and the significance of measuring pH, temperature, and FT-IR spectra at different time intervals.
#12 The differences in raw material protein source and formulation of the products are mentioned, but a more detailed comparison of carbohydrate profiles and protein content would be valuable. Specifically, how these differences impact the fermentation process could be briefly discussed.
#13 The FT-IR measurement section is detailed, including the instrument used, spectral range, and data preprocessing steps. However, it would be beneficial to briefly mention the rationale for selecting FT-IR as a technique and its advantages in this context.
#14 The HPLC sampling and analysis section provides detailed information on the quantification of lactic acid. However, a brief mention of why these specific analytes (glucose, fructose, and lactic acid) were chosen for spiking experiments would enhance the reader's understanding.
#15 The multivariate data analysis section is comprehensive, providing details on data preprocessing, modeling, and evaluation. However, a brief explanation of why certain spectral regions were removed and the rationale behind choosing specific variables for analysis would enhance clarity.
#16 When referencing techniques such as mean centering, Savitszky-Golay smoothing, and multivariate data analysis, consider providing appropriate references for readers who may want to delve deeper into these methods.
In results and discussion
#17 The section is comprehensive but could benefit from a more explicit structure. Consider subheadings for each major point discussed, making it easier for readers to follow.
#18 The inclusion of Figures 2, 3, and 4 is beneficial. However, providing more detailed captions that explain specific trends or observations in the figures would be helpful.
#19 Clearly stating how outliers were identified and handled is crucial. If outliers were removed, provide information on the criteria used for outlier detection and the impact of their removal on the results.
#20 While the results are presented in detail, more interpretation of the observed trends is needed. Discuss the implications of the findings on the overall fermentation process and product quality.
#21 Discuss the significance of lactic acid concentrations in the context of product quality and consumer acceptance. Provide comparisons with known ranges in similar products, as mentioned in the literature.
#22 Clarify how compositional differences (protein and carbohydrate content) contribute to the observed spectral variations. Discuss the potential impact of these differences on product attributes.
#23 The discussion on spectral changes during fermentation is insightful. Elaborate on how these changes correlate with the ongoing biochemical processes, specifically relating to pH variations.
#24 The PCA analysis is valuable, but further explanation is needed. Discuss the biological relevance of principal components and how they correlate with specific chemical changes or processes during fermentation.
#25 While the pH prediction model is discussed, provide additional insights into how well the model performed during different phases of fermentation. Discuss the practical implications of these models for real-time monitoring.
General Comments
#26 Please rewrite the abstract and conclusion after making the necessary changes.
Comments on the Quality of English Language
Please check the report!
Author Response
Please find our point to point responses in the attached file

Reviewer 2 Report
Comments and Suggestions for Authors
The study focuses on developing quality monitoring methods for fermenting pea and oat-based yogurt-like products using Fourier Transform Infrared (FT-IR) spectroscopy and High-Performance Liquid Chromatography (HPLC). It demonstrates that FT-IR spectroscopy can effectively track fermentation variations based on pH and lactic acid with high predictive accuracy, offering a faster and reliable alternative to traditional methods. The main variance during fermentation was linked to shifts in spectral carbohydrate and protein bands.
1.The study is limited to pea and oat-based products. Additional research on a broader range of plant-based substrates could enhance the applicability of the findings.
2. While the study provides RMSEP and R2 values, further external validation with independent datasets could strengthen confidence in the predictive models.
3.The study simplifies a complex fermentation process to spectral changes in carbohydrates and proteins. It might overlook other critical biochemical changes occurring during fermentation that could impact product quality.
4.The paper could benefit from a more detailed comparison with existing fermentation monitoring methods, highlighting specific advantages and limitations of FT-IR spectroscopy in real-world applications. PLS is a latent variable model for fundementally linear regression analysis. However, modern machine learning can boost the regressor for application.
5.The study does not extensively address how variations in fermentation conditions (e.g., temperature, starter culture concentration) might affect the spectral data and the accuracy of predictions.
Minors:
1.Some technical descriptions of the spectroscopy and HPLC methods could be elaborated for clarity and reproducibility.
2.Additional statistical analyses could be provided to support the robustness of the findings, especially concerning outlier detection and the handling of spectral data.
3.Suggestions for future research directions could be more explicitly stated, particularly in applying these methods to industrial-scale fermentation monitoring.
4.Some represented figures are too vague (eg. Fig 6).
Author Response
Please find our point-to-point responses to reviewer in the attached file

Round 2
Reviewer 1 Report
Comments and Suggestions for Authors
Thank you by your responses. The manuscript can be now accepted.
Reviewer 2 Report
Comments and Suggestions for Authors